# Cross-Species Transmission of Rabbit Hepatitis E Virus to Pigs and Evaluation of the Protection of a Virus-like Particle Vaccine against Rabbit Hepatitis E Virus Infection in Pigs

**DOI:** 10.3390/vaccines10071053

**Published:** 2022-06-30

**Authors:** Sang-Hoon Han, Hee-Seop Ahn, Hyeon-Jeong Go, Dong-Hwi Kim, Da-Yoon Kim, Jae-Hyeong Kim, Kyu-Beom Lim, Joong-Bok Lee, Seung-Yong Park, Chang-Seon Song, Sang-Won Lee, Yang-Kyu Choi, In-Soo Choi

**Affiliations:** 1Department of Infectious Diseases, College of Veterinary Medicine, Konkuk University, 120 Neungdong-ro, Gwangjin-gu, Seoul 05029, Korea; silverxi@konkuk.ac.kr (S.-H.H.); frequency0@konkuk.ac.kr (H.-S.A.); goluffy@konkuk.ac.kr (H.-J.G.); opeean0@konkuk.ac.kr (D.-H.K.); kimda68@konkuk.ac.kr (D.-Y.K.); mirine2u@konkuk.ac.kr (J.-H.K.); 4316rbqja2@konkuk.ac.kr (K.-B.L.); virus@konkuk.ac.kr (J.-B.L.); paseyo@konkuk.ac.kr (S.-Y.P.); songcs@konkuk.ac.kr (C.-S.S.); odssey@konkuk.ac.kr (S.-W.L.); 2Department of Laboratory Animal Medicine, College of Veterinary Medicine, Konkuk University, 120 Neungdong-ro, Gwangjin-gu, Seoul 05029, Korea; yangkyuc@konkuk.ac.kr; 3KU Center for Animal Blood Medical Science, Konkuk University, 120 Neungdong-ro, Gwangjin-gu, Seoul 05029, Korea; 4Konkuk University Zoonotic Diseases Research Center, Konkuk University, 120 Neungdong-ro, Gwangjin-gu, Seoul 05029, Korea

**Keywords:** hepatitis E virus, rabbit, swine, cross transmission, cross protection, VLP vaccine, fibrosis, inflammation

## Abstract

We investigated the cross-species transmission of rabbit hepatitis E virus (rb HEV) to pigs and evaluated the cross-protection of a swine (sw) HEV-3 virus-like particle (VLP) vaccine against rb HEV infection in pigs. Twelve 4-week-old conventional pigs were divided into negative control (*n* = 3), positive control (rb HEV-infected, *n* = 4), and vaccinated (vaccinated and rb HEV-challenged, *n* = 5) groups. The vaccine was administered at weeks 0 and 2, and viral challenge was conducted at week 4. Serum HEV RNA, anti-HEV antibody, cytokine, and liver enzyme levels were determined. Histopathological lesions were examined in abdominal organs. Viral RNA was detected and increased anti-HEV antibody and alanine aminotransferase (ALT) levels were observed in positive control pigs; liver fibrosis, inflammatory cell infiltration in the lamina propria of the small intestine and shortened small intestine villi were also observed. In vaccinated pigs, anti-HEV antibody and Th1 cytokine level elevations were observed after the second vaccination; viral RNA was not detected, and ALT level elevations were not observed. The results verified the cross-species transmission of rb HEV to pigs and cross-protection of the sw HEV-3 VLP vaccine against rb HEV infection in pigs. This vaccine may be used for cross-protection against HEV infection in other species.

## 1. Introduction

Hepatitis E virus (HEV), a quasi-enveloped single-stranded positive-sense RNA virus, is a member of the family *Hepeviridae*, which is composed of two genera, *orthohepevirus* and *pischihepevirus*. The genus *orthohepevirus* is divided into four species from A to D. HEV in *orthohepevirus* A has one serotype and has been classified into eight genotypes: HEV-1 to HEV-8 [1]. HEV-1 and HEV-2 only infect humans and are transmitted through the fecal–oral route in developing countries [2,3]. HEV-3 and HEV4 have been isolated from humans and several animal species, including pigs. As zoonotic viruses, HEV-3, HEV-4, and HEV-7 cause sporadic and autochthonous hepatitis E mainly in developed countries [3]. Humans can be infected with these viruses by consuming raw or undercooked animal meats and products [3,4,5], and these viruses show cross-species transmission from other animal species to humans [6,7].

In Korea, the prevalence and seroprevalence of HEV in humans were reported to be 2.64% and 11.9–17.7%, respectively [8,9].

HEV infection is a major public health concern. HEV infection causes self-limiting acute hepatitis [10]. The mortality rate of HEV infection is <1%. However, in HEV-infected pregnant women in developing countries, such as India, the mortality rate can be as high as 20% or higher. Furthermore, severe complications, such as fulminant hepatic failure and adverse fetal outcomes, including abortion, stillbirth, and intrauterine death, can occur [11,12,13,14]. Moreover, chronic hepatitis occurs in immunocompromised individuals, such as patients with human immunodeficiency virus-1 infections and hematological malignancies and in solid organ transplant recipients [15,16,17,18].

Pigs are the primary animals responsible for zoonotic HEV infections. There are many reports of zoonotic transmission from pigs to humans in individuals who eat raw or undercooked meat, xenotransplantation patients, veterinarians, and farmers who work with swine [19,20,21]. Swine (sw) HEV is genetically similar to human HEV. The sw HEV gene encoding open reading frame 2 (ORF2) shares 79–80% identity at the nucleotide level and 90–92% identity at the amino acid level with human HEV sequences [22]. Pigs are the main hosts for many HEV strains and are thus a major experimental animal model for HEV studies [23,24,25].

Rabbits have also been identified as HEV hosts. In 2009, rabbit (rb) HEV was first isolated from farmed Chinese rabbits [26]. It was demonstrated that rb HEVs were genetically closely related to human and sw HEV-3 but distant from other HEV genotypes based on phylogenetic analysis of the complete genome [27,28,29]. Furthermore, many studies have reported the cross-species transmission of rb HEV from rabbits to other animal species, including humans [30,31,32,33]. Pigs and rabbits are recognized as important animals for HEV research.

Several reports have indicated that ribavirin may be effective in the treatment of severe acute and chronic HEV-3 infection [34,35]. However, the prevention of HEV infection is also important because cross-species transmission of HEV can occur. Because it is difficult to produce HEV in cell-culture systems, vaccine development has mainly been focused on the virus-like particle (VLP) composed of capsid proteins [36,37,38,39]. Currently, only one VLP vaccine, which is composed of 239 amino acids (368–606 amino acids) of the capsid protein, has been approved for humans in China [39]. The 239-aa VLP induces neutralizing antibodies, providing good protection in humans [40]. However, this vaccine was constructed using the capsid protein of only HEV-1, and further development of vaccines with other HEV genotypes is necessary. Furthermore, the broad use of vaccines based on HEV-3 has been suggested because HEV-3 is the most prevalent genotype and a major causative agent of HEV infection across the world.

Cross-species transmission is a major issue in HEV research, and multilateral studies are required to elucidate its mechanism and thus prevent HEV cross-species transmission. In this study, cross-species transmission of rb HEV in pigs and the efficacy of cross-protection of the sw HEV-3 VLP vaccine against rb HEV were evaluated.

## 2. Materials and Methods

### 2.1. Experimental Animals

All the animal experiments were approved by the Institutional Animal Care and Use Committee of Konkuk University (IACUC No. KU21240). Twelve 4-week-old, conventional female Large Yorkshire pigs were obtained from pig farms in Korea. Pigs in a non-HEV-infected state were confirmed using a commercial anti-HEV antibody enzyme-linked immunosorbent assay (ELISA) kit (Wantai, Beijing, China) and nested reverse transcription-polymerase chain reaction (RT-PCR) using gene-specific primers for HEV-3 and HEV-4, as previously described [41]. Pigs were freely fed with tap water and feed (Sunjin, Seoul, Korea). The animal’s room was maintained at a temperature of 23 ± 3 °C with 50 ± 10% humidity and illuminated repeatedly at 150–300 Lux at 12 h intervals. Pigs were adapted for at least 1 week before the experiment. Pigs were divided into the following three experimental groups: negative control (mock-infected, *n* = 3), positive control (rb-HEV-infected, *n* = 4), and vaccine (vaccinated against sw HEV-3 and challenged with rb-HEV, *n* = 5) groups.

### 2.2. Virus and Vaccines

Rb HEV was obtained from rabbit fecal samples and used to inoculate pigs in the positive control and vaccinated groups. Fecal samples were suspended in a 10-fold volume of phosphate-buffered saline (PBS; pH 7.4) and centrifuged at 3000× *g* for 30 min. The supernatant was collected, filtered through a 0.2 mm syringe filter, and stored at −70 °C until use. The virus was confirmed using nested RT-PCR and sequencing analysis using gene-specific primers, as previously described [27].

The genomic equivalents (GE) of rb HEV were determined using quantitative RT-PCR with gene-specific primer sets, as previously described [42]. The viral titer of rb-HEV used for the challenge was adjusted to 10^5^ GE/mL in 2% bovine albumin solution (BSA). 1X PBS was injected in the mock group.

The vaccine used for the evaluation of cross-species protection was based on the ORF2 region of sw HEV-3, as described in a previous study [43]. The VLP was composed of 239 amino acids (368–606) of HEV-3 isolated from pigs in Korea (Appendix A). The 239-aa VLP, which is a part of HEV ORF2, is known to be highly immunogenic and safe [40,44,45]. The sw HEV-3 VLP vaccine (200 µg) was administered intramuscularly at 0 and 2 weeks based on previous studies in our laboratory.

### 2.3. RNA Extraction from Fecal and Serum Samples

Fecal and serum samples were collected from all pigs weekly for 10 weeks after vaccination. Fecal samples were suspended in PBS (1:10) and centrifuged at 3000× *g* for 30 min, and the supernatants were collected. Blood samples were collected using a Vacutainer^®®^ (BD Biosciences, San Jose, CA, USA) and centrifuged at 3000× *g* for 20 min to obtain sera. All fecal and serum samples were stored at −70 °C until use. Viral RNA was extracted from 150 μL of serum or fecal samples using the Patho Gene-spin DNA/RNA kit, according to the manufacturer’s instructions (Intron, Gyeonggi-do, Korea). RNA samples were stored at −70 °C.

### 2.4. Detection of the Partial HEV Genomic Sequence

Nested RT-PCR was performed to detect the partial genomic sequence of HEV ORF2, using gene-specific primers. The final PCR products were identified using gel electrophoresis. The PCR products were extracted from the agarose gel and cloned into a TA cloning vector (RBC^TM^, New Taipei City, Taiwan). The cloned vectors were transformed into HIT-competent Cells^TM^ DH5a (RBC^TM^, New Taipei City, Taiwan). Sequencing analysis was performed using plasmid DNA extracted from the colonies selected using a blue-white screen.

### 2.5. Anti-HEV Antibody Detection

The titers of anti-HEV antibodies in serum samples collected over 10 weeks were determined using a commercial ELISA kit (Wantai, Beijing, China) according to the manufacturer’s instructions.

### 2.6. Measurement of ALT and Aspartate Aminotransferase (AST) Levels

ALT and AST levels in serum samples were measured using a UV assay according to the procedure of the International Federation of Clinical Chemistry and Laboratory Medicine without pyridoxal phosphate activation (Neodin, Seoul, Korea).

### 2.7. Measurement of Cytokine Levels

The levels of interleukin (IL)-1b, tumor necrosis factor (TNF)-a, IL-10, IL-4, IL-12, and interferon (IFN)-γ in serum samples were measured using commercial ELISA kits (R&D Systems, Minneapolis, MN, USA) according to the manufacturer’s instructions. 

### 2.8. Histopathology

Abdominal tissue samples from the liver, spleen, mesenteric lymph node (M.L), small intestine (S.I), and large intestine (L.I) were fixed using 10% neutral buffered formalin and embedded in paraffin. The same tissue areas were collected and analyzed from each pig. Tissues were stained with hematoxylin and eosin (H&E) to identify inflammatory cells and with Masson’s trichrome stain to identify fibrosis. The ratio of the area of fibrotic lesions to the total area of liver tissue and the height of S.I villi were measured using MetaMorph^®®^ (Molecular Devices, San Jose, CA, USA).

### 2.9. Statistical Analysis

The data were analyzed using a two-way analysis of variance (ANOVA) in the GraphPad Prism software (version 8.0.2; GraphPad Software, San Diego, CA, USA). Dunnett’s multiple comparison test was performed as a post hoc analysis. A *p*-value of <0.05 was considered statistically significant.

## 3. Results

### 3.1. Detection of HEV RNA

HEV RNA in fecal samples and serum samples was detected using nested RT-PCR. As expected, no viral RNA was detected in the fecal or serum samples collected from negative control pigs (Table 1). In the positive control group, viral RNA in fecal and serum samples was confirmed in three out of four pigs. Viral RNA was first detected in serum samples 2 weeks post-infection (wpi), and viral shedding was confirmed until 5 wpi (Table 1). However, viral RNA was not detected in the fecal or serum samples collected from vaccinated pigs (Table 1). These results indicate that rb HEV could infect pigs across the species barrier and that the sw HEV-3 VLP vaccine could establish cross-protection against rb HEV in pigs.

### 3.2. Determination of the Anti-HEV Antibody Levels

The presence of anti-HEV antibodies in serum samples was detected using ELISA. As expected, no anti-HEV antibodies were detected in negative control pigs (Figure 1). In contrast, seroconversion was confirmed in serum samples from positive control pigs and vaccinated pigs (Figure 1). In the vaccinated group, the titer of anti-HEV antibodies in serum samples was initially elevated 1 week after the first vaccination, and the titer elevation lasted until 1 week after the second vaccination (Figure 1). Elevation of the anti-HEV antibody titer was observed in both vaccinated pigs and positive control pigs at 2 wpi. However, the anti-HEV antibody titer in the vaccinated group was significantly higher than that in the positive control group (*p* < 0.001) (Figure 1). These results indicated that the sw HEV-3 VLP vaccine induced the production of anti-HEV antibodies and memory B cells. Furthermore, the immunity elicited by sw HEV-3 VLP vaccines could protect against HEV infection.

### 3.3. Determination of Cytokine Levels

The serum levels of IL-12 and IFN-γ, representing Th1 immune responses, IL-4 and IL-10, representing Th2 immune responses, and IL-1β and TNF-α, representing pro-inflammatory cytokines were measured at 0, 2, 4, 6, 8, and 10 weeks. The levels of cytokines for the Th2 immune response and those of pro-inflammatory cytokines in the serum samples from vaccinated and positive control pigs were not significantly changed compared to those from negative control pigs (Figure 2A–D). However, the serum levels of IL-12 and IFN-γ, which represent Th1 immune responses in vaccinated pigs, were significantly elevated compared with those in negative and positive control pigs (*p* < 0.001) (Figure 2E,F). A statistically significant increase in the serum levels of IL-12 was observed 2 weeks after the second vaccination. Elevation of IFN-γ levels in serum samples was observed 2 weeks after the second vaccination. Furthermore, a significant increase was confirmed compared to that in the negative and positive control groups at 4 weeks after the second vaccination, although there was a slight decrease compared to that observed at 2 weeks after the second vaccination (Figure 2E,F).

### 3.4. Determination of Liver Enzyme Levels

ALT and AST levels in the serum samples of all pigs were analyzed at 0, 2, 4, 6, 8, and 10 weeks. As expected, there was no elevation in ALT and AST levels in the serum samples from negative control pigs (Figure 3A,B). AST levels in the serum samples were also not increased in positive control pigs (Figure 3B). However, serum ALT levels increased in the positive control group from 1 wpi until 6 wpi (*p* < 0.05) (Figure 3A). In contrast, the elevation of ALT and AST levels was not observed in the serum samples from vaccinated pigs (Figure 3A,B). These results indicate that cross-species rb HEV infection and cross-protection of the sw HEV-3 VLP vaccine against rb HEV were successfully established in pigs. 

### 3.5. Histopathological Examination

To identify histopathologic lesions in abdominal organs such as the liver, spleen, S.I, L.I, and M.L, H&E staining, and Masson’s trichrome staining were conducted. The infiltration of inflammatory cells was confirmed by H&E staining, while fibrosis was confirmed by Masson’s trichrome staining. There was no infiltration of inflammatory cells, and occasional fibrotic lesions were observed in the liver tissues of the negative control pigs, as expected (Figure 4A and Figure 5A). In the positive control group, there were no significant differences in inflammatory lesions in liver tissues (Figure 4B). However, severe fibrosis was observed in the liver tissues of positive control pigs (Figure 5B). Similar to the negative control group, infiltration of inflammatory cells in the liver was not detected, and only occasional fibrotic lesions were identified in the liver tissues of vaccinated pigs (Figure 4C and Figure 5C). On average, the area of fibrosis was approximately 4.6% in the negative control group and 5% in the vaccinated group. The positive control group demonstrated significantly higher (*p* < 0.001) fibrosis levels (16.6%) than the negative and vaccinated groups (Figure 5D). Infiltration of inflammatory cells in the lamina propria of S.I of the positive control pigs was higher than that of the negative control pigs (Figure 6A,B). However, in vaccinated pigs, infiltration of inflammatory cells was observed in the lamina propria of the S.I to a degree similar to that of the negative control pigs (Figure 6C). When the height of the S.I villi was measured, the average lengths were significantly higher in the negative control (211 mm) and vaccinated (207.2 mm) pigs than in positive control (132.7 mm) pigs (*p* < 0.001) (Figure 6D). In M. L., the number of lymphoid follicles and germinal centers seemed to be higher in positive control and vaccinated pigs than that in negative control pigs (Appendix A). However, in the spleen, there were no significant differences in inflammatory cell infiltration and the formation of germinal centers among the three groups (Appendix A). When the inflammatory lesions of L.I were compared, no difference was observed between the three groups (Appendix A).

## 4. Discussion

In developing countries in Asia and Africa, HEV-1 and HEV-2 are often water-transmitted, whereas HEV infections in developed countries are zoonotic and are mainly caused by the transmission of HEV-3 and HEV-4 due to consumption of raw or undercooked meats [2,46,47,48]. Various studies on the pathogenesis and immune response to HEV have been conducted, including studies in various experimental animals [23,49,50,51,52]. Non-primate experimental animal models, such as chimpanzee, cynomolgus monkey, and rhesus monkey models, have many advantages in HEV studies. These animals are susceptible to a wide range of genotypes, including HEV-1, HEV-2, HEV-3, and HEV-4. Furthermore, they show tropism for viral replication and pathogenic consequences similar to humans [53]. However, there are some limitations to HEV research because these animal models are not easy to access owing to limited facilities and costs. Recently, cross-species transmission of HEV has emerged as a major concern, and the representative genotypes for cross-species transmission are HEV-3 and HEV-4. Therefore, HEV-3 and HEV-4 infections and cross-species transmission of HEV are major topics in HEV studies, in which pigs and rabbits are commonly used. Various experiments on HEV infection from other species have been carried out in pigs and rabbits [23,43,52,54]. Cross-species transmission of sw HEV-3 in rabbits was conducted in a previous study, and cross-species transmission of rb HEV in pigs and cross-protection of the sw HEV-3 VLP vaccine were identified in this study [52].

Rb HEV is phylogenetically similar to HEV-3 and can be classified as a HEV-3 variant [27]. Therefore, it can be inferred that there is a possibility of HEV cross-infection in pigs and rabbits. As expected, cross-species transmission of rb HEV in pigs has been successfully established. Approximately 75% of pigs were confirmed to be infected with rb HEV, and viremia, fecal shedding, and seroconversion were detected. According to the results of viral replication and shedding, cross-species transmission of rb HEV in pigs was more established than that of sw HEV-3 in rabbits, and it was also confirmed that the duration of the infection was longer in the cross-species transmission of rb HEV in pigs than in the cross-transmission of sw HEV-3 in rabbits [52]. Anti-HEV antibody production in serum was identified at 2 wpi, and the antibody titer was elevated continuously until 6 wpi. Furthermore, the serum ALT levels increased, which is one of the major indices of HEV infection. In HEV studies using experimental animals, there are some cases in which the levels of liver enzymes do not increase during HEV infection [52,55,56,57]. However, in this study, a significant increase in the ALT levels was observed in the positive control group of pigs infected with rb-HEV, which could indicate that cross-species transmission was established. Furthermore, our results indicate a more definite infection than that reported in a previous study of sw HEV-3 cross-species infection in rabbits [52].

HEV has one serotype, and rb-HEV and HEV-3 are closely related genetically, so there is a possibility of cross-protection between HEV-3 and rb-HEV [1,27,58]. Cross-protection of pigs infected with rb-HEV was achieved in this study using the sw HEV-3 VLP vaccine. After administration of the HEV vaccine, the pattern of changes in anti-HEV antibody production varied according to that in previous studies [43,56]. The VLP vaccine used in this study caused seroconversion 1 week after the first vaccination. In addition, the anti-HEV antibody titer increased rapidly at 2 wpi in the vaccinated group. These results implied that humoral immunity against HEV was clearly established and acted in response to rb-HEV infection. In addition, cytokine levels in the serum were confirmed to reflect the activation of the immune response. It was confirmed that the levels of IL-12 and IFN-γ, representative cytokines related to the Th1 immune response, were elevated. Two weeks after the second vaccination, the IL-12 and IFN- γ levels were significantly increased compared to those of the negative control, and 4 weeks after the second vaccination, the IFN-γ levels were slightly decreased but significantly higher than those of the negative control group. These results indicate that the sw HEV-3 VLP vaccine sufficiently stimulated the Th1 immune response. Th1 immune responses play an important role in the clearance of other hepatitis viruses [59,60]. Therefore, these results indicate that the sw HEV-3 VLP vaccine efficiently stimulated both humoral and cellular immunity. However, there was no elevation in Th1 cytokine levels in the serum of the positive control group. ORF3 plays a major role in HEV immune evasion. It is presumed that the VLP vaccine produced using only the capsid protein ORF2 sufficiently stimulated immunity, while the whole virus evaded immunity based on ORF3 action [61]. Several studies on cross-protection against HEV infection have shown effective protection; however, the rate of non-responders was approximately 30% [58,62]. However, the cross-protection of the sw HEV-3 VLP vaccine against rb HEV in this study was elicited successfully, as there was no viral replication, viral shedding, or elevation of ALT.

Despite the obvious evidence of HEV infection, such as viral replication, viral shedding, and elevation of ALT in serum samples from positive control pigs, there was no elevation in pro-inflammatory cytokine levels. Several studies have offered different opinions on the correlation between cytokines and ALT, which also varies depending on the cause of hepatitis [63,64,65,66]. Additional studies on the correlation between cytokine and ALT levels in HEV infection are needed. Histological examination revealed no inflammatory lesions in the liver tissues of positive control pigs. These results might be due to an insufficient virus titer to cause inflammatory lesions or the result of analysis after recovery. Severe histopathological lesions were associated with the virus titer in human HEV and rb HEV infections in a recent study [67]. In addition, it has been reported that inflammatory lesions in the liver disappear over time after infection [57]. Several studies reported that HEV infection causes fibrotic lesions in the liver [43,52]. Similarly, in this study, higher levels of fibrosis were identified in the liver of positive control pigs than those in negative control and vaccinated pigs. Several studies have shown that HEV infects not only the liver but also other organs, such as the S.I, L.I, lymph nodes, and uterus [68,69,70]. In addition, it has been reported that hepatitis B causes not only infection but also inflammation in the intestine [71]. According to a recent study, enterocytes infected with HEV strongly increase the secretion of thymic stromal lymphopoietin, which induces inflammation and tissue injury during infection [72]. In addition, HEV infection with aspergillosis causes abruption and edema of the intestinal villi in Himalayan Griffons [70]. In this study, infiltration of inflammatory cells in the lamina propria of S.I and significantly decreased height of villi in S.I were identified in positive control pigs compared with those in negative control and vaccinated pigs. These results suggest that cross-species transmission of rb HEV to pigs might not be limited to the liver but could also occur in the S.I with histopathological lesions. The developed sw HEV-3 VLP vaccine is successfully cross-protected against rb HEV infection in pigs. These results indicate that the VLP vaccine would protect HEV from transmitting across animal species and would thus be applicable to prevent HEV infection in several animal species and humans.

## 5. Conclusions

In conclusion, this study demonstrates that cross-species transmission of rb-HEV in pigs can occur. These results suggest that cross-species transmission might occur in different species of animals that are susceptible to HEV-3 and rb HEV, including humans. In addition, the sw HEV-3 VLP vaccine efficiently stimulated humoral and cellular immunity and successfully cross-protected against rb HEV in pigs. The results of this study contribute to the understanding of the pathogenesis of HEV cross-species transmission. They are useful for understanding cross-protection and preventing the spread of HEV between different animal species.

## Figures and Tables

**Figure 1 vaccines-10-01053-f001:**
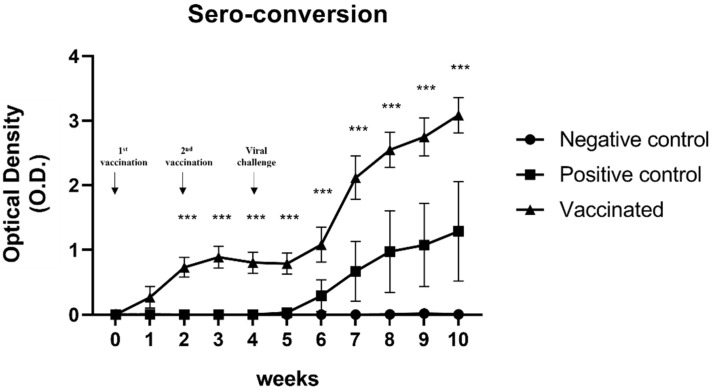
Determination of the anti-hepatitis E virus (HEV) antibody titers: Titers of anti-HEV antibodies in the serum were measured using ELISA. Anti-HEV antibody titers were elevated 1 week after vaccination in the vaccinated group and 2 weeks after the viral challenge in the positive control group. Statistical significance was determined between groups at * *p* < 0.05, ** *p* < 0.01, and *** *p* < 0.001.

**Figure 2 vaccines-10-01053-f002:**
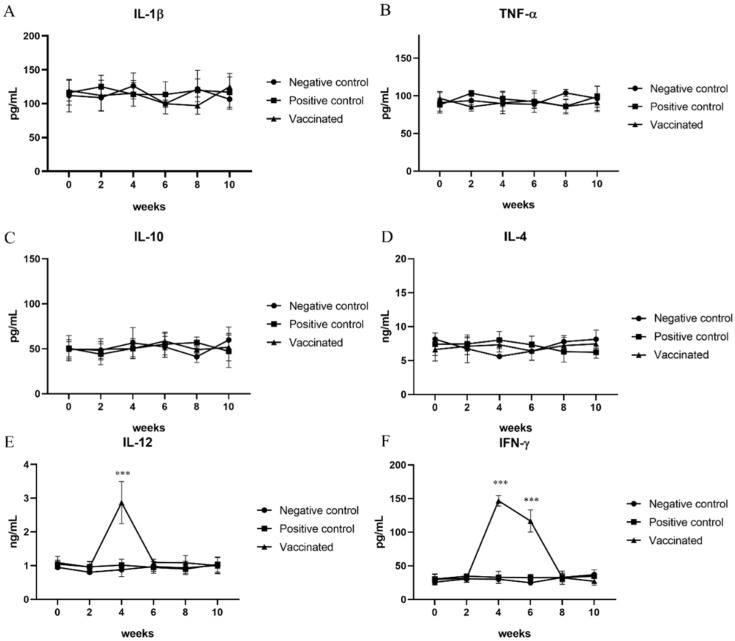
Determination of cytokine levels: To study the immune response to hepatitis E virus infection, the levels of (**A**) interleukin (IL)-1β; (**B**) tumor necrosis factor (TNF)-α; (**C**) IL-10; (**D**) IL-4; (**E**) IL-12; and (**F**) interferon (IFN)-γ were measured using ELISA every other week. IL-12 and IFN-γ represent the Th1 immune response, which was elevated after vaccination. Statistical significance was determined between groups at * *p* < 0.05, ** *p* < 0.01, and *** *p* < 0.001.

**Figure 3 vaccines-10-01053-f003:**
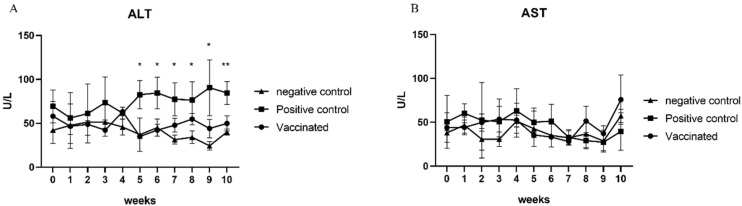
Determination of hepatic enzyme levels: alanine aminotransferase (ALT) (**A**) and aspartate aminotransferase (AST) (**B**) levels were measured every other week. There were no significant changes in AST levels. However, a significant elevation in ALT levels was detected in the vaccinated group. Statistical significance was determined between groups at * *p* < 0.05, ** *p* < 0.01, and *** *p* < 0.001.

**Figure 4 vaccines-10-01053-f004:**
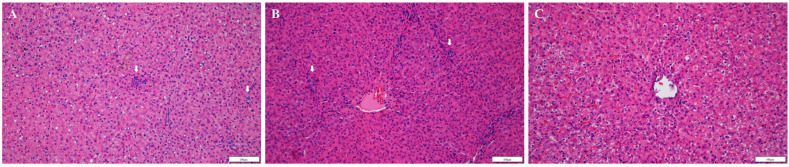
Detection of inflammatory lesions in the liver: tissue sections (**A**–**C**) were stained with hematoxylin and eosin stain; (**A**) negative control pigs; (**B**) positive control pigs; (**C**) vaccinated group pigs. Arrows indicate infiltration of inflammatory cells.

**Figure 5 vaccines-10-01053-f005:**
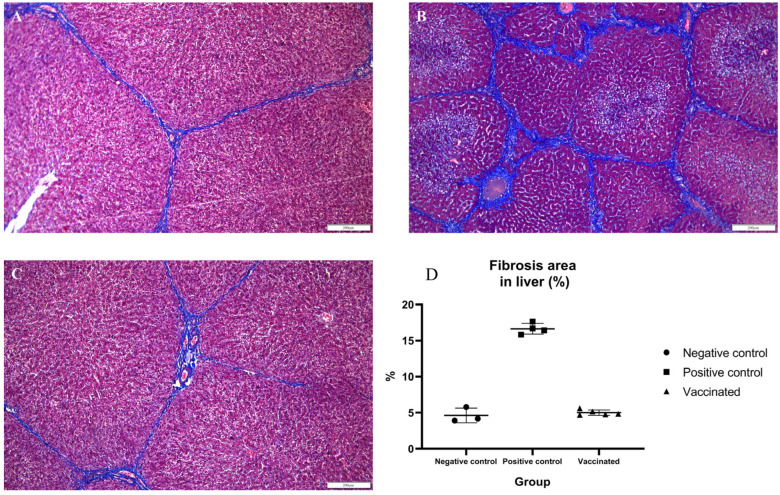
Detection of fibrotic lesions in the liver: tissue sections (**A**–**C**) were stained with Masson’s trichrome stain; (**A**) negative control pigs; (**B**) positive control pigs; (**C**) vaccinated group pigs; (**D**) statistical analysis of fibrotic area in the liver.

**Figure 6 vaccines-10-01053-f006:**
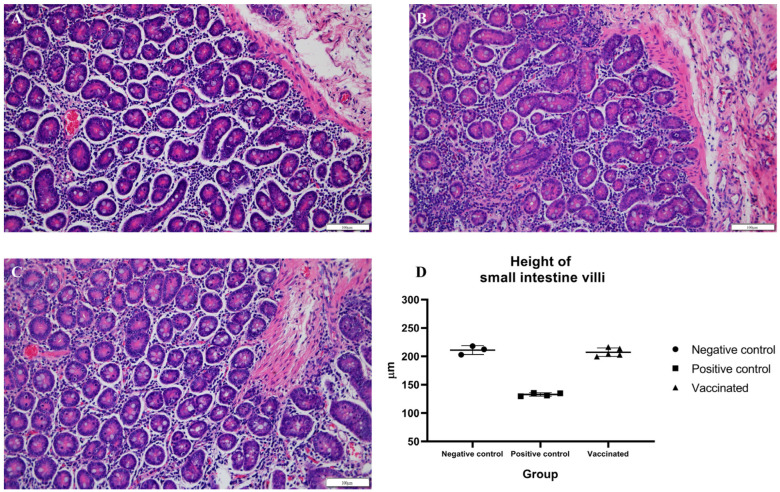
Detection of histopathological lesions in the small intestine: tissue sections (**A**–**C**) were stained with hematoxylin and eosin stain; (**A**) negative control pigs; (**B**) positive control pigs; (**C**) vaccinated group pigs; and (**D**) height of small intestine villi.

**Table 1 vaccines-10-01053-t001:** HEV detection in fecal and serum samples. ^a^ F/S: feces/serum.

Group	Week	0	1	2	3	4	5	6	7	8	9	10
	No.	^a^ F/S	F/S	F/S	F/S	F/S	F/S	F/S	F/S	F/S	F/S	F/S
Negative Control	1	−/−	−/−	−/−	−/−	−/−	−/−	−/−	−/−	−/−	−/−	−/−
	2	−/−	−/−	−/−	−/−	−/−	−/−	−/−	−/−	−/−	−/−	−/−
	3	−/−	−/−	−/−	−/−	−/−	−/−	−/−	−/−	−/−	−/−	−/−
Positive Control	4	−/−	−/−	−/−	−/−	−/−	−/−	−/−	−/−	−/−	−/−	−/−
	5	−/−	−/−	−/−	−/−	−/−	−/−	−/+	+/+	+/+	+/−	−/−
	6	−/−	−/−	−/−	−/−	−/−	−/−	−/+	−/+	+/+	+/+	−/−
	7	−/−	−/−	−/−	−/−	−/−	−/−	−/−	+/+	+/+	+/−	−/−
Vaccinated	8	−/−	−/−	−/−	−/−	−/−	−/−	−/−	−/−	−/−	−/−	−/−
	9	−/−	−/−	−/−	−/−	−/−	−/−	−/−	−/−	−/−	−/−	−/−
	10	−/−	−/−	−/−	−/−	−/−	−/−	−/−	−/−	−/−	−/−	−/−
	11	−/−	−/−	−/−	−/−	−/−	−/−	−/−	−/−	−/−	−/−	−/−
	12	−/−	−/−	−/−	−/−	−/−	−/−	−/−	−/−	−/−	−/−	−/−

## Data Availability

The data are contained within the article and Appendix A.

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
