# Peer review of "Cross-Species Transmission of Rabbit Hepatitis E Virus to Pigs and Evaluation of the Protection of a Virus-like Particle Vaccine against Rabbit Hepatitis E Virus Infection in Pigs"

_vaccines, 2022, doi:10.3390/vaccines10071053_

Round 1

Reviewer 1 Report

Authors reported “Cross-species transmission of rabbit hepatitis E virus to pigs 2

and evaluation of the protection of a virus-like particle vaccine 3 against rabbit hepatitis E virus infection in pigs.” Why did not use the swine HEV? Authors should describe the reason.

1.      Authors should describe the parts of HEV for VLP vaccine in detail and show them using one figure.

2.      Authors should describe the Swine authors used and the condition of keeping in detail.

3.      How the prevalence of HEV infection in your country?

4.      Authors should discuss how to use this vaccine in the real-world.

5.      In introduction section, “…countries[2]. HEV-3 and HEV4 have been isolated from humans and animals, including pigs and wild boars. Humans can be infected with HEV-3 or HEV-4 by consuming raw or undercooked meat[3, 4]. HEV…” See “HEV-1 and HEV-2 are restricted to humans and associated with outbreaks in developing countries where the virus is transmitted through the fecal–oral route, while HEV-3 and HEV-4 are zoonotic with an expanded host range and are the main cause of sporadic and autochthonous cases of hepatitis E in developed countries. Zoonotic cases caused by HEV-3 and HEV-4 strains are mostly associated with strains from pigs and wild boars.” In the following reference: Primadharsini PP, Nagashima S, Okamoto H. Genetic Variability and Evolution of Hepatitis E Virus. Viruses. 2019 May 18;11(5):456. doi: 10.3390/v11050456. PMID: 31109076

6.      See: Kamar N, et al. Ribavirin for chronic hepatitis E virus infection in transplant recipients. N Engl J Med. 2014 Mar 20;370(12):1111-20. doi: 10.1056/NEJMoa1215246.

PMID: 24645943. All the patients had HEV viremia when ribavirin was initiated (all 54 in whom genotyping was performed had HEV genotype 3). Authors should extensively revise the introduction section.

Author Response

Point 1: Why did not use the swine HEV? Authors should describe the reason.

 Response 1: We already conducted the experiment showing protective effects on swine HEV with swine VLP vaccine and the results were published in journal vaccines (Go et al., 2021). The purpose of this study was to establish cross-species transmission and evaluate cross-protection of VLP vaccine against rabbit HEV.

 Point 2: Authors should describe the parts of HEV for VLP vaccine in detail and show them using one figure.

 Response 2: page 3, lines 120-123. A detailed description of the VLP vaccine used in this study has been described in a previous study (Go et al., 2021). We added the description of HEV for VLP vaccine in Materials and method section as you requested. The VLP was composed of 239 amino acids (368-606) in HEV-3 isolated from pig in Korea. The 239 VLP, a part of HEV ORF2, is known to be highly immunogenic and safe (Zhu et al., 2010; Wu et al., 2012; Li et al., 2005). We provided a supplementary figure showing the position of VLP sequence in HEV genome as you requested.

 Point 3: Authors should describe the swine used and the condition of keeping in detail.

Response 3: page 3, lines 102-105. We added the animal care conditions in Materials and method section as you requested. A total of 12, four-week-old, conventional female Large Yorkshire pigs were obtained from pig farms in Korea. Pigs were freely fed with tap water and feed (Sunjin, Korea). The conditions of the animal room were maintained at a temperature of 23±3℃ with 50±10% humidity and illuminated repeatedly at 150-300 Lux with 12 h intervals. Pigs were adapted for at least 1 week prior to the experiment.

 Point 4: How the prevalence of HEV infection in your country?

Response 4: page 2, lines 75-77. We added the prevalence of HEV infection in Korea in introduction section as you requested. In Korea, seroprevalence of HEV in human was about 17.7%(Choi et al., 2003). In pigs and SPF rabbits, the seroprevalence of HEV were 14.8% and 4%, respectively(Choi et al., 2003; Han et al., 2018).

 Point 5: Authors should discuss how to use this vaccine in the real-world.

Response 5: page 10, lines 355-358. We added the discussion how to use this vaccine in the real-world in discussion section as you requested. The developed sw HEV-3 VLP vaccine successfully cross-protected rb HEV infection in pigs. These results indicate that the VLP vaccine would protect HEV transmission cross animal species. This vaccine would be applicable to prevent HEV infection in several animal species and humans.

Point 6: In introduction section, “…countries [2]. HEV-3 and HEV4 have been isolated from humans and animals, including pigs and wild boars. Humans can be infected with HEV-3 or HEV-4 by consuming raw or undercooked meat[3, 4]. HEV…” See “HEV-1 and HEV-2 are restricted to humans and associated with outbreaks in developing countries where the virus is transmitted through the fecal–oral route, while HEV-3 and HEV-4 are zoonotic with an expanded host range and are the main cause of sporadic and autochthonous cases of hepatitis E in developed countries. Zoonotic cases caused by HEV-3 and HEV-4 strains are mostly associated with strains from pigs and wild boars.” In the following reference: Primadharsini PP, Nagashima S, Okamoto H. Genetic Variability and Evolution of Hepatitis E Virus. Viruses. 2019 May 18;11(5):456. doi: 10.3390/v11050456. PMID: 31109076

Response 6: page 1-2, lines 45-51. We changed the contents of manuscript more specifically and added the reference as you suggested.

Point 7: See: Kamar N, et al. Ribavirin for chronic hepatitis E virus infection in transplant recipients. N Engl J Med. 2014 Mar 20;370(12):1111-20. doi: 10.1056/NEJMoa1215246.

PMID: 24645943. All the patients had HEV viremia when ribavirin was initiated (all 54 in whom genotyping was performed had HEV genotype 3). Authors should extensively revise the introduction section.

 Response 7: page 2, lines 78-89. Information of HEV treatment using ribavirin and development of HEV vaccines is added in introduction as you suggested.

Reviewer 2 Report

This is very interesting work about Virus like particles and their efficacy in mounting 

1. It would be great if the author shows that VLP vaccines could mount immune responses in rabbits against rb HEV since this would also prevent the virus transmission in human and also the VLP vaccine is not species specific.

2. It would be important to show that the immune response mounted is rb HEV specific since the VLP vaccine could mount immune responses against other viruses as well.

3. What was injected in the mock group? Is it just 1XPBS or other VLPs? It is necessary check whether the immune responses are similar for other VLPs. 

4. Table 1 week 10, all the groups showed absence of Virus. Does this mean the Virus in fecal and serum samples are cleared without vaccines?

5. Line 167, the memory cells are B or T cells? Since both can be necessary for the protection.

6. Are the pigs re-infected with rb HEV or HEV 3 to check efficacy of the VLP vaccine?

7. The cytokines IL-12 and IFN-y level was detected in vaccinated group, but is it HEV-3 VLP specific? Also, it would be nice if the serum detection can be done for individual pigs since one pigs serum cytokine level can affect the reading.

8. Also, it would be useful to check other body fluids for the presence of viruses since VLP may affect virus migration in the body.

9. The antibodies are detected using VLP vaccine, but it would be nice to show whether those antibodies could neutralize viruses.

10. Was the reduced ALT level due to vaccine effect?

11. Figure 4, the pictures of histopathologic sections need arrow indicators for inflammatory cells.

12. Figure 5 & 6, the author needs to mention whether similar areas have been analyzed for liver and small intestine from different groups since different areas could show variable pathologies.

13. Figure 5 & 6, it would be nice to put the titles on the X-axis in the graphs for liver fibrosis and small intestine villi height instead of on the side of the graph.

Author Response

Point 1: It would be great if the author shows that VLP vaccines could mount immune responses in rabbits against rb HEV since this would also prevent the virus transmission in human and also the VLP vaccine is not species specific.

 Response 1: We did the experiments showing the VLP effects on rabbit HEV in a separate study. We did identify the complete protection of rabbit HEV infection in rabbits by the VLP. Those results are submitted to other journal and currently under review. Therefore, we could not include those results in this study. Please, understand our situations.

Point 2: It would be important to show that the immune response mounted is rb HEV specific since the VLP vaccine could mount immune responses against other viruses as well.

Response 2: As you know, it is known that HEV has only one serotype [1]. The 239 VLP vaccine includes a neutralizing antibody epitope. In this study, we used a commercial ELISA kit which can detect anti-HEV antibodies produced from any animals including rabbits. In figure 1, seroconversions were observed only after vaccination with VLP and viral challenge in pigs. Therefore, these results indicated that VLP vaccines elicit rabbit HEV-specific immune response. 

Point 3: What was injected in the mock group? Is it just 1XPBS or other VLPs? It is necessary check whether the immune responses are similar for other VLPs. 

Response 3: page 3, line 118. 1XPBS was injected in pigs of mock group.

 Point 4: Table 1, week 10, all the groups showed absence of virus. Does this mean the virus in fecal and serum samples are cleared without vaccines?

Response 4:  Yes, you are right. Because HEV causes an acute infection in animals, the virus typically appears at 1-6 weeks in fecal and serum samples after viral challenge. After that period, virus are not detected from specimens of pigs including positive control group.

Point 5: Line 167, the memory cells are B or T cells? Since both can be necessary for the protection.

Response 5: page 5, line 188. We suppose memory B cells are involved in the production of anti-HEV antibodies after vaccination. We did not evaluated T-cell mediated cellular immune response in this study.

Point 6: Are the pigs re-infected with rb HEV or HEV 3 to check efficacy of the VLP vaccine?

Response 6: Pigs in positive control group and in the vaccinated group were challenged with rabbit HEV at 4 weeks. Therefore, we evaluated the infectivity of HEV in positive control and the protective efficacy of VLP vaccine in the vaccinated group.

Point 7: The cytokines IL-12 and IFN-y level was detected in vaccinated group, but is it HEV-3 VLP specific? Also, it would be nice if the serum detection can be done for individual pigs since one pigs serum cytokine level can affect the reading.

Response 7: We collected serum samples from pigs at week 4 prior to viral challenge. Therefore, we are sure the increase of IL-12 and IFN-g were induced by two times of VLP vaccination. In fig 2, we presented the mean concentration of cytokines in all pigs in each group. We provide a supplementary table providing IL-12 and IFN-g levels of each pig.

Point 8: Also, it would be useful to check other body fluids for the presence of viruses since VLP may affect virus migration in the body.

Response 8: We are sorry. We did not check virus in other body fluids in this study. In a further study, we will determine the presence of HEV in other body fluids such as saliva.

Point 9: The antibodies are detected using VLP vaccine, but it would be nice to show whether those antibodies could neutralize viruses.

Response 9: It makes better to show whether antibodies could neutralize virus as you mentioned. However, we could not confirm whether antibodies neutralize virus or not because we did not develop a cell culture system for HEV in our laboratory. Please, understand our situations.

Point 10: Was the reduced ALT level due to vaccine effect?

Response 10: ALT levels were not elevated in vaccinated pigs after viral challenge compared to those of positive control pigs (p<0.05).

Point 11: Figure 4, the pictures of histopathologic sections need arrow indicators for inflammatory cells.

Response 11: As you mentioned, arrows indicating inflammatory cells were added in figure 4.

Point 12: Figure 5 & 6, the author needs to mention whether similar areas have been analyzed for liver and small intestine from different groups since different areas could show variable pathologies.

Response 12: page 4, lines 155-156. We analyzed the same areas of the organs of all pigs without bias.

 Point 13: Figure 5 & 6, it would be nice to put the titles on the X-axis in the graphs for liver fibrosis and small intestine villi height instead of on the side of the graph.

Response 13: We added the titles on the X-axis in fig 5 and 6 as you suggested.

Round 2

Reviewer 1 Report

This reviewer has several minor comments.

1. "In Korea, the seroprevalence of HEV in humans was reported to be 17.7% [30]. In pigs and SPF rabbits, the seroprevalence of HEV was reported to be 14.8% and 4%, respectively [30, 31]." Authors should show the prevalence of HEV in Korea.

2.  I do not agree with the opinions from authors " Response 1: We already conducted the experiment showing protective effects on swine HEV with swine VLP vaccine and the results were published in journal vaccines (Go et al., 2021). The purpose of this study was to establish cross-species transmission and evaluate cross-protection of VLP vaccine against rabbit HEV." in the responses from authors.

Authors should describe this in this paper and refer the references: 

Small Animal Models of Hepatitis E Virus Infection. Li TC, Wakita T. Cold Spring Harb Perspect Med. 2019 Aug 1;9(8):a032581. doi: 10.1101/cshperspect.a032581. PMID: 29735581

Characterization of rabbit hepatitis E virus isolated from a feral rabbit. Mendoza MV, Yonemitsu K, Ishijima K, Minami S, Supriyono, Tran NTB, Kuroda Y, Tatemoto K, Inoue Y, Okada A, Shimoda H, Kuwata R, Takano A, Abe S, Okabe K, Ami Y, Zhang W, Li TC, Maeda K. Vet Microbiol. 2021 Dec;263:109275. doi: 10.1016/j.vetmic.2021.109275. Epub 2021 Nov 11. PMID: 34798367

Author Response

Response to Reviewer 1 Comments

Point 1: "In Korea, the seroprevalence of HEV in humans was reported to be 17.7% [30]. In pigs and SPF rabbits, the seroprevalence of HEV was reported to be 14.8% and 4%, respectively [30, 31]." Authors should show the prevalence of HEV in Korea.

Response 1: Page 2, Line 52-53. We added the prevalence of HEV infection in Korea in introduction section as you requested and provided the reference [8, 9]. We changed the sentence like this “In Korea, the prevalence and seroprevalence of HEV in humans was reported to be 2.64% and 11.9-17.7%, respectively.”

 We deleted the following sentences because they are not relevant to the main contents of the manuscript. “In pigs and SPF rabbits, the seroprevalence of HEV was reported to be 14.8% and 4%, respectively.”

 Point 2: I do not agree with the opinions from authors " Response 1: We already conducted the experiment showing protective effects on swine HEV with swine VLP vaccine and the results were published in journal vaccines (Go et al., 2021). The purpose of this study was to establish cross-species transmission and evaluate cross-protection of VLP vaccine against rabbit HEV." in the responses from authors.

Authors should describe this in this paper and refer the references: 

Small Animal Models of Hepatitis E Virus Infection. Li TC, Wakita T. Cold Spring Harb Perspect Med. 2019 Aug 1;9((Li and Wakita 2019)

Characterization of rabbit hepatitis E virus isolated from a feral rabbit. Mendoza MV, Yonemitsu K, Ishijima K, Minami S, Supriyono, Tran NTB, Kuroda Y, Tatemoto K, Inoue Y, Okada A, Shimoda H, Kuwata R, Takano A, Abe S, Okabe K, Ami Y, Zhang W, Li TC, Maeda K. Vet Microbiol. 2021 Dec;263:109275. doi: 10.1016/j.vetmic.2021.109275. Epub 2021 Nov 11. PMID: 34798367

 Response 2: page 2, lines 71-73. We additionally described the characteristics of rabbit HEV and included the two references as you suggested [27-29]. We think this information would be enough for supporting why we used rabbit HEV in pigs to demonstrate cross-species transmission of rabbit HEV.

Reviewer 2 Report

It seems author has made all the changes in the publication as suggested.

Author Response

Thank you for your revision.